# AXL Receptor Tyrosine Kinase as a Promising Therapeutic Target Directing Multiple Aspects of Cancer Progression and Metastasis

**DOI:** 10.3390/cancers14030466

**Published:** 2022-01-18

**Authors:** Marie-Anne Goyette, Jean-François Côté

**Affiliations:** 1Montreal Clinical Research Institute (IRCM), Montréal, QC H2W 1R7, Canada; marie-anne.goyette@ircm.qc.ca; 2Molecular Biology Programs, Université de Montréal, Montréal, QC H3T 1J4, Canada; 3Department of Biochemistry and Molecular Medicine, Université de Montréal, Montréal, QC H3C 3J7, Canada; 4Department of Anatomy and Cell Biology, McGill University, Montréal, QC H3A 0C7, Canada

**Keywords:** AXL, metastasis, invasion, EMT, tumor microenvironment, fibrosis, angiogenesis, immune evasion, hypoxia

## Abstract

**Simple Summary:**

Metastasis is a complex process that requires the acquisition of certain traits by cancer cells as well as the cooperation of several non-neoplastic cells that populate the stroma. Cancer-related deaths are predominantly associated with complications arising from metastases. Limiting metastasis therefore represents an important clinical challenge. The receptor tyrosine kinase AXL is required at many steps of the metastatic cascade and contributes to tumor microenvironment deregulation. In this review, we describe how AXL contributes to metastatic progression by governing various biological processes in cancer cells and in stromal cells, highlighting the potential of its inhibition.

**Abstract:**

The receptor tyrosine kinase AXL is emerging as a key player in tumor progression and metastasis and its expression correlates with poor survival in a plethora of cancers. While studies have shown the benefits of AXL inhibition for the treatment of metastatic cancers, additional roles for AXL in cancer progression are still being explored. This review discusses recent advances in understanding AXL’s functions in different tumor compartments including cancer, vascular, and immune cells. AXL is required at multiple steps of the metastatic cascade where its activation in cancer cells leads to EMT, invasion, survival, proliferation and therapy resistance. AXL activation in cancer cells and various stromal cells also results in tumor microenvironment deregulation, leading to modulation of angiogenesis, fibrosis, immune response and hypoxia. A better understanding of AXL’s role in these processes could lead to new therapeutic approaches that would benefit patients suffering from metastatic diseases.

## 1. Introduction

AXL is a member of the TAM family of receptor tyrosine kinases (RTKs) that also includes TYRO3 and MER. This was the last family of RTKs to be identified and because their inactivation in mice resulted in rather mild phenotypes, their biological roles were slow to be characterized. Now, however, in addition to being recognized as key regulators of immune cell activation, TAM RTKs have been shown to be expressed in cancer cells where they promote survival and invasion and contribute to resistance to various therapies. The structure shared by these RTKs includes an extracellular domain consisting of tandem repeats of immunoglobulin-like and fibronectin type 3 (FN-III)-like domains, a single-pass transmembrane domain and an intracellular domain that includes a catalytically competent kinase defined by a unique KWIAIES conserved sequence [1]. While a few genetic mutations and amplifications have been reported in TAMs in cancers (listed in [2]), their functional importance has yet to be defined.

Nevertheless, it is now well established that AXL expression is linked to increased risk of metastasis and poor survival in a variety of solid cancers including breast cancer, non-small cell lung carcinoma, ovarian cancer, and clear cell renal carcinoma [3,4,5,6,7,8]. AXL is expressed and plays a role in different types of cells including tumor cells, fibroblasts, vascular cells and several immune cells. Thus, in a cancer context, AXL expression in tumor cells and in stromal cells can contribute to the progression of the disease. This review highlights novel discoveries about AXL functions in different tumor compartments and discusses potential therapeutic interventions through AXL inhibition.

## 2. AXL Activation and Therapeutic Targeting

AXL can be activated in various ways and these activation mechanisms are unique. Thus, several therapeutic targeting strategies have been designed and evaluated to efficiently inhibit AXL.

### 2.1. Ligand-Dependent and Ligand-Independent Mechanisms of Activation

The principal ligands that lead to TAM activation are growth arrest specific factor 6 (GAS6) and protein S, both of which require vitamin-K dependent γ-carboxylation to achieve maximal activation [9,10]. These ligands vary in their affinity for the different TAM receptors: GAS6 binds all TAM receptors, with the highest affinity for AXL, while protein S only binds MER and TYRO3. These ligands can also bind the lipid moiety of phosphatidylserine (PS) and can activate TAMs when exposed on apoptotic cells, aggregating platelets, exosomes or virus envelopes [11,12,13]. It has also been shown that GAS6 activation of AXL is localized at regions with high GAS6 concentration, resulting in a diffusional influx of AXL and receptor aggregation and dimerization [14]. Furthermore, GAS6 can mediate dimer formation of other TAM members (MER or TYRO3), which have a lower affinity for GAS6 binding, and more experiments are needed to determine if the receptors can also heterodimerize and how this affects their signaling [15,16,17]. Many laboratory studies have shown that the GAS6/AXL axis promotes cell invasion, proliferation and survival, thus contributing to cancer progression and metastasis. Conversely, clinical studies have shown a correlation between GAS6 expression and patients’ survival in breast cancer, suggesting that GAS6 may not be essential in this context [18,19]. In support of this, it was observed that *GAS6* is dispensable in a pre-clinical model of HER2^+^ breast cancer for the formation of metastases [3]. Interestingly, AXL can be activated in a ligand-independent manner in many pathological contexts. For example, its increased expression, or the presence of oxidative stress, can lead to AXL homodimerization and autophosphorylation [20,21,22]. Furthermore, AXL has been shown to crosstalk and heterodimerize or cluster with other RTKs such as EGFR, HER2, HER3, MET, PDGFR and VEGFR-2 to promote downstream signaling [3,23,24,25,26,27,28,29]. These partnerships serve to diversify the downstream signaling of these RTKs and confer advantages to cancer cells. Through such a mechanism, AXL can promote resistance to a variety of therapies, including chemotherapy and targeted therapies including inhibitors of its partners EGFR, HER2 and PDGF [23,30,31,32,33,34,35,36].

### 2.2. AXL Therapeutic Targeting

AXL is a particularly interesting candidate as a therapeutic target because its genetic deletion or pharmacological inhibition in mice is well tolerated. Various approaches to inhibit AXL have been studied for cancer treatments including small molecule inhibitors that compete with ATP-binding or monoclonal antibodies [37,38,39]. Indeed, a number of clinical trials are ongoing with the AXL inhibitor R428 (also known as BGB324 or Bemcentinib), which is highly specific for AXL among the TAMs and other RTKs [39]. Additionally, a number of other broad spectrum kinase inhibitors that would also inhibit AXL are currently under study and have been reviewed in [40]. Other approaches have also been envisioned to inhibit the GAS6/AXL axis, including decoy receptors that trap GAS6 and Vitamin K antagonists that would reduce the ability of GAS6 to activate AXL [41,42,43]. However, because there are multiple ways to activate AXL as previously mentioned, there may not be an absolute requirement for GAS6 in certain contexts. More recently, an original approach exploited AXL as a cancer antigen for chimeric antigen receptor (CAR)-T cell therapy. In triple-negative breast cancer (TNBC) where AXL is overexpressed, engineered T cells with AXL-CAR-T were able to induce cytokine release and an antigen-specific cytotoxicity [44,45].

Before we can apply AXL-targeting therapeutic strategies to their full potential in cancer treatment, it is important to better understand the consequences of effective, systemic AXL inhibition. As such, a more detailed understanding of the functions of AXL in various tumor cell types, including both in cancer and different stromal cells, is needed. Accordingly, the next sections will discuss our current understanding of AXL function in a variety of cancer-related biological processes that are both intrinsic and extrinsic to cancer cells.

## 3. Cancer Cell Intrinsic Implications of AXL Expression in the Metastatic Cascade and Therapy Resistance

Increased AXL expression has been shown to correlate with decreased patients’ survival and metastasis of cancer cells in a plethora of solid cancers. Several biological triggers within the tumors that are known to promote metastasis, such as TGF-β/EMT and hypoxia, impact AXL expression levels in cancer cells [3,4,46,47,48]. Signals downstream of AXL advantage cancer cells throughout the metastatic process and lead to therapy resistance. AXL has been implicated in many steps of the metastatic cascade and recent studies have clarified the cancer cell-intrinsic AXL contributions to this complex process (Figure 1).

### 3.1. Epithelial-to-Mesenchymal Transition (EMT)

An emerging function of AXL during cancer progression is its role in the plastic and dynamic program called epithelial-to-mesenchymal transition (EMT). EMT affords several advantages to tumor cells by enhancing cell invasion, apoptosis resistance and stem-like characteristics at the expense of proliferation, therefore contributing to tumor progression and therapy resistance [49]. AXL expression correlates strongly with EMT markers in various cancers [50]. It is now well established that EMT induces the expression of AXL via different EMT transcription factors like SNAIL, SLUG, TWIST or ZEB2 [4,46]. As a feedback mechanism, AXL activation then upregulates the expression of EMT transcription factors and mesenchymal markers to sustain EMT [38,51,52]. Thus, AXL plays multiple roles in tumor progression and metastasis through its modulation of several steps of the metastatic cascade and therapy resistance.

### 3.2. Cell Invasion and Migration

Invasive characteristics of cancer cells are particularly important in the first steps of the metastatic cascade including invasion of the tumor parenchyma and transendothelial migration during the entry (intravasation) and exit (extravasation) of the systemic circulation. In addition to the role of AXL in enhancing cell invasion by promoting mesenchymal features, other elements triggered by AXL signaling are also important to promote cell motility. For example, AXL can activate several generic downstream effectors from such RTKs as MAPK, PI3K/AKT and JAK/STAT to promote actin reorganization, migration and survival of cancer cells [53,54,55,56,57,58]. Indeed, AXL has been shown to regulate cell adhesion via NCK2/ILK and this signaling converges on the integrin β3 pathway to stimulate cell adhesion [59,60]. In addition, AXL was shown to phosphorylate ELMO proteins in complex with the GEF DOCK1, leading to RAC-mediated cytoskeleton remodeling and migration of TNBC cells [61]. More recently, the first phosphoproteome of AXL was generated following stimulation of AXL by GAS6 in TNBC cells to uncover specific AXL effectors. This phosphoscreen identified NEDD9/CRKII/PEAK1 downstream of AXL and described a novel role for AXL in focal adhesion turnover [62]. This screen also suggested the existence of many other processes and pathways affected by AXL signaling such as RNA transport and vesicle trafficking. Additional research is required to determine which of these AXL-mediated signaling events contributes to invasion and metastasis.

AXL has also been identified as a direct HIF target gene in clear cell renal cell carcinoma (ccRCC) [47]. During hypoxic stress, ccRCC cells upregulate AXL to promote metastasis by maximizing invasion via the GAS6/AXL signaling cascade leading to MET activation. AXL was also shown to regulate HIF-1α levels, thus contributing to the hypoxic response in HER2^+^ breast cancer cells [48]. In this context, AXL is required for hypoxia-induced EMT and invasion leading to metastasis.

In human cell lines, AXL expression seems to be restricted to invasive and mesenchymal lines, raising the possibility that AXL is a marker for mesenchymal cancers, like TNBC [63]. Nevertheless, its expression in breast cancer correlates with patients’ survival and metastasis independently of the subtype [3]. For example, AXL was found to correlate with metastasis in HER2^+^ breast cancer, a subtype that retains epithelial characteristics even though it often leads to metastasis and evidence suggests that EMT is required for this capacity [3,64]. Indeed, during progression of epithelial cancer, only a small population of motile cells invade the surrounding tissue to disseminate and form metastases. TGF-β, a factor that induces EMT, is required in several steps of the metastatic process including local invasion, intravasation and extravasation [65,66,67]. Interestingly, in epithelial cell lines that do not express AXL at basal levels, like HER2^+^ cells and patient-derived xenografts (PDX), TGF-β reprogramming can promote AXL expression [3,68]. In this context, AXL deletion in a preclinical model of HER2^+^ breast cancer almost completely abrogated the metastatic dissemination by blocking intravasation, extravasation and growth at the metastatic site [3], supporting a role for AXL in TGF-β-induced cell invasion.

These new findings regarding AXL signaling, and biology further confirm its role in cancer cell dissemination through the reprogramming of cells to be able to leave the primary tumor and enter a distant organ by increasing cell motility.

### 3.3. Modulation of the Metastatic Niche and Dormancy

Following entry in a distant organ, cells must adapt to the foreign environment to form macrometastases. To do so, mesenchymal cells must be reprogrammed to a more epithelial state to be able to proliferate. In a HER2^+^ breast cancer model, inducible AXL downregulation after the arrival of cancer cells in the lungs led to a reduction of metastatic outgrowth [3]. A similar observation was made using cells isolated from the PyMT murine breast cancer model and with MDA-MB-231 breast cancer cells where it was found that AXL inhibition post-seeding reduced the metastatic burden [69]. By using AXL^+^ tumor initiating cells that display partial EMT, it was suggested that AXL prepared the metastatic niche by activating lung fibroblasts via the secretion of THSB2. Those fibroblasts subsequently promoted a switch toward a more epithelial state, reducing AXL levels and promoting proliferation of cancer cells leading to the formation of macrometastasis.

Disseminated cancer cells can also enter a dormant state in the new organ instead of proliferating and can then survive in a quiescent state for years where they are highly resistant to treatment and can lead to relapses in patients [70]. Both AXL and GAS6 have been implicated in dormancy in the context of bone marrow metastasis of prostate cancer. In the bone marrow niche, osteoblasts produce GAS6 that can activate AXL on the surface of nearby disseminated cancer cells [71,72]. AXL activation then induces the expression of TGF-β2 and TGF-βRs that results in the arrest of proliferation and cell survival [72]. AXL has also been reported to be strongly expressed in dormant myeloma cells and its inhibition induces their proliferation [73]. Altogether, these results suggest that the persistence of AXL expression and signaling in disseminated cancer cells can lead to dormancy in the metastatic niche in some cancers.

### 3.4. Therapy Resistance

AXL promotes and sustains a mesenchymal phenotype leading to metastasis, but its role in EMT also has an important impact on therapy outcome. Indeed, EMT is associated with a wide range of changes linked to stemness and drug resistance, implicating AXL as an important mediator of resistance to chemotherapy, antimitotic drugs, and various targeted therapies.

In support of this role, AXL has been found to be upregulated in chemo-resistant cells in a variety of cancers. Many studies have shown a better drug sensitivity when combining AXL inhibition with chemotherapeutic compounds such as docetaxel, cisplatin, pemetrexel, vincristine, paclitaxel, adryamicin, gemcitabine or carboplatin [74,75,76,77,78]. In cancer cells, chemotherapy induces EMT and AXL is also upregulated in this context [51,56,68]. Indeed, AXL was found to be associated with mesenchymal features of breast and lung cancer cells and its inhibition synergized with antimitotic agents to induce cell death [68]. Furthermore, genetic, or pharmacologic inhibition of AXL was shown to revert EMT in pancreatic and prostate cancer cells and to modulate the expression of nucleoside transporters playing a role in chemotherapeutic response [75,77]. Additionally, many studies link resistance to the EGFR inhibitor Erlotinib to EMT and AXL overexpression [57,79]. In this context, erlotinib-resistant cells displayed EMT features that were prevented by AXL inhibition. However, EMT-associated resistance to erlotinib has also been reported to be independent of AXL [68] and further studies are needed to explain this discrepancy. Crizotinib, an ALK inhibitor, has also been linked to the activation of AXL and induction of EMT [80]. Thus, interfering with AXL can improve responses to various therapies by reverting EMT and dampening downstream pathways that lead to drug resistance. In conclusion, numerous studies provide rationale for the use of AXL inhibition to enhance sensitivity and/or prevent resistance to cytotoxic chemotherapies and targeted agents, highlighting the potential of AXL-targeted drugs in combination with a variety of other therapies.

## 4. AXL as a Modulator of the Tumor Microenvironment

AXL is also known to contribute to cancer progression, metastasis and therapy resistance by reprogramming the tumor microenvironment to be more favorable for cancer progression. Indeed, AXL is expressed by several non-cancerous cells including endothelial cells, fibroblasts, monocytes, platelets, natural killer cells (NK), dendritic cells (DC), and macrophages. Therefore, AXL activation in cancer cells and stromal cells could also promote disease progression by acting on angiogenesis, fibrosis, immune evasion, and hypoxia (Figure 2).

### 4.1. Angiogenesis

Primary tumor growth and metastatic progression depend on the ability of cancer cells to promote angiogenesis to deliver nutrients and oxygen to cells in the tumor. Some cancer cells have the capacity to secrete proangiogenic factors that will recruit endothelial cells and degrade the extracellular matrix, thereby contributing to abnormal angiogenesis in the tumor. This angiogenesis then leads to tumor and metastatic growth and AXL has been shown to play a role in this process. Indeed, AXL downregulation or inhibition in cancer cells reduces the secretion of angiogenic factors leading to suppression of the angiogenic capacity of breast cancer cells both in vitro and in vivo [81]. Furthermore, in ccRCC cells, AXL expression correlates with antiangiogenic resistance, mediated by the GAS6/AXL/S100A10 axis promoting plasmin production, endothelial cell migration, and angiogenesis [82].

AXL and GAS6 are expressed by different components of the cardiovascular system including endothelial cells, pericytes and vascular smooth muscle cells (VSMC). The GAS6/AXL axis has been implicated in vasculogenesis by affecting the migration and apoptosis of vascular smooth muscle cells (VSMC) [83]. Furthermore, autocrine, and paracrine GAS6/AXL signaling is required for endothelial tube formation in vitro, suggesting a role for AXL in angiogenesis [84,85,86]. In pericytes, paracrine activation of AXL by GAS6 was shown to regulate the expression of the pro-angiogenic protein Cyr61, leading to angiogenesis and tumor growth [87]. Thus, several studies now support the notion that AXL inhibition in vivo could lead to a reduction in tumor angiogenesis or a normalization of the blood vessels [39,48,81,82].

### 4.2. Fibrosis

In the context of liver fibrosis, the soluble extracellular domain of AXL is cleaved and released into the circulation where it is an accurate biomarker to detect advanced liver fibrosis and cirrhosis [88]. AXL is involved in the development of fibrosis, linking AXL with non-alcoholic fatty liver disease (NAFLD)/non-alcoholic steatohepatitis (NASH) and hepatocellular carcinoma (HCC) [89]. Indeed, advanced liver fibrosis characterized by inflammation, injury, and hepatic fibrosis predisposes patients to HCC, suggesting that AXL can be an oncogenic driver in this context [90,91]. The AXL/GAS6 axis is required for the activation of hepatic stellate cells (HSC) that can promote a permissive environment for cancer development by the production of extracellular matrix and inflammation [89,90]. In mice, interfering with AXL functions reduced HSC activation, liver fibrosis and inflammation, and prevented development of NASH [89].

### 4.3. Generation of an Immunosuppressive Microenvironment

A dampened immune response is permissive for tumor progression. In some cases, cancer cells have been shown to interact with immune cells to reduce their anti-tumoral activity and highjack them to support their progression. To achieve this, certain cancer cells either secrete chemokines that modulate the behaviour and recruitment of immune cells or modulate the expression of molecules at their cell surface that serve as communication modules with the immune system to increase or decrease their activity. Several studies now suggest that AXL contributes to an immunosuppressive tumor microenvironment and can modulate the ability of immune cells to eliminate cancer cells.

Indeed, AXL has been reported to upregulate the secretion of immunosuppressive cytokines such as CSF1-3, CCL2-5, CXCL1,2,5, IL-1a, IL-6, TGF-β and TNF-α [48,77,92,93]. AXL signaling has also been associated with immunosuppressive macrophages in a cancer context where conditioned media from AXL expressing cancer cells cause macrophages to polarize in vitro [48,94,95]. AXL can also contribute to immune escape by decreasing major histocompatibility complex class I (MHC-I) molecules present at the cell surface, leading to a decrease in antigen presentation, which is important for the antitumor immune response [92,93]. AXL has also been reported to drive the expression of the immune checkpoint protein PD-L1 on cancer cells via the PI3K-AKT axis to prevent T-cell activation [92,93,96,97]. Experimentally, inhibiting AXL increases the recruitment and activation of T cells and dendritic cells and decreases the infiltration and polarization of macrophages toward a pro-tumoral phenotype [48,92,93]. Interestingly, a recent study also reported a role of TAMs in promoting the oligomerization of MLKL, leading to necroptosis, a pro-inflammatory necrotic cell death [98], underscoring the biological complexity of these receptors.

Within the immune system, TAMs are primarily expressed by antigen presenting cells (macrophages and dendritic cells) and NK cells. TAM receptors have crucial roles in innate immunity. Indeed, the triple KO mice of TAMs are viable but suffer from chronic inflammation and systemic autoimmunity [99]. TAMs are in the center of a negative feedback loop that prevents overactivation of adaptive immunity and serves to maintain homeostasis. They reduce inflammation by different mechanisms including the phagocytosis of apoptotic cells, the regulation of toll-like receptor (TLR) signaling and the production of cytokines and interferons by macrophages and dendritic cells [15,100,101,102,103,104]. Unlike in cancer cells, the downstream effectors of AXL signaling leading to cytokine secretion in immune cells have been well characterized. Indeed, TLRs induce the expression of pro-inflammatory cytokines and their activation also upregulates AXL. The upregulation of AXL triggers a negative feedback loop via STAT1/SOCS1-3, inhibiting TLR signaling to maintain tissue homeostasis [102]. Therefore, AXL inhibition could stimulate antitumor immunity by modifying the production of pro-inflammatory factors by immune cells.

Many studies have also described roles for the TAMs in immune cells in a tumor setting. For example, the TAMs have been shown to act as a break for NK cell activation by inhibiting the downstream signaling of the activating receptor NKG2D [42]. The degradation of TAMs via the E3 ubiquitin ligase Cbl-b, or their pharmacological inhibition, induces a robust NK cell anti-metastatic activity. The GAS6/AXL axis has also been implicated in regulatory T cell (Tregs) suppressive activity, where AXL activation by GAS6 on CD4^+^CD25^+^ Tregs leads to an increase in their immunosuppressive activity in vitro and in vivo [105]. Additionally, TAMs are upregulated in tumor-derived myeloid suppressor cells (MDSC) where they are required for immunosuppressive functions [106]. Furthermore, AXL was also found to upregulate the expression of PD-L1 in dendritic cells, influencing T cell infiltration and activation in tumors [107]. These recent observations suggest that AXL can affect the antitumor immune response in a great variety of ways and that AXL inhibition could have positive impacts at several levels.

### 4.4. Hypoxia

Hypoxia resulting from rapid tumor growth is another important component of the tumor microenvironment and can be amplified by feedback mechanisms from surrounding tissues. Hypoxia generates a response that leads to deregulation of cancer cell behaviour and interaction with the stroma by modulating the expression of genes related to angiogenesis, immune evasion and invasion [108]. Importantly, AXL was recently shown to be required to generate the hypoxic response in HER2^+^ breast cancer by regulating HIF-1α expression [48]. AXL was found to be essential for cancer cell hypoxia-induced EMT, invasion and cytokine production. These, in turn, led to macrophage proliferation, invasion, and polarization, thus deregulating the tumor microenvironment, and promoting metastasis. These studies suggest that AXL inhibition in a preclinical murine model considerably modulates the tumor microenvironment by changing the immune infiltration and normalizing the blood vessels leading to increased perfusion and decreased leakiness.

### 4.5. Therapeutic Implications

A body of work now suggests that inhibition of AXL, and in some cases TAMs more generally, leads to an antitumorigenic microenvironment by affecting angiogenesis and immune response at the cancer and stromal cells levels. These data explain why systemic inhibition of AXL is emerging as a promising approach to enhance anti-angiogenic therapies and immunotherapies.

Since AXL is implicated in tumor angiogenesis, its role in acquired resistance to angiogenesis inhibitor therapy has been of great interest. Sunitinib, an anti-angiogenic small molecule, was shown to increase AXL expression and signaling [109,110]. Chronic treatment with Sunitinib induces AXL and MET RTKs, leading to invasion and angiogenesis. In this context, co-inhibition of AXL and MET impaired Sunitinib-acquired resistance in vitro and in vivo [109]. Thus, a combination of inhibitors could provide an effective approach to counter resistance to angiogenesis inhibitors.

Cancer immunotherapy using checkpoint inhibitors is another emerging option in the clinic, but many patients are unresponsive. Mechanisms of evasion include adaptive immune resistance, where tumor cells promote an immunosuppressive environment leading to T cell exclusion and inactivation [111]. Since AXL inhibition has been shown to promote a pro-inflammatory microenvironment, normalize blood vessels, and reduce immune evasion and hypoxia, combining it with immune checkpoint blockade or radiation could be a viable solution to increase their efficacy. For example, while a combination of radiation therapy and checkpoint immunotherapy have been suggested to treat various cancers, tumors resistant to this strategy have been shown to overexpress AXL [92]. In these tumors, AXL inhibition increased the sensitivity of combination therapy by increasing the CD8+ T cell response. Furthermore, since AXL activation is linked to PD-L1 expression on cancer cells, combining PD-L1 and AXL inhibition was able to increase the anti-tumor efficacy in vivo [48,93,112]. In addition, broad-spectrum inhibition of TAMs has been reported to enhance the effect of immune checkpoint blockage [106,113]. In conclusion, numerous lines of investigation identify AXL as a strong therapeutic target with the aim to enhance the efficacy of cancer treatments.

## 5. Future Directions

In recent years, AXL has been implicated in a variety of biological processes leading to tumor progression, metastasis, and resistance to therapies. Indeed, its expression is linked with EMT, invasion, metastasis and changes in the tumor microenvironment including hypoxia, angiogenesis, and immune evasion. Thus, AXL is an attractive therapeutic target to impair multiple stages of cancer progression by affecting a wide variety of cell types within the tumor. Nevertheless, a number of important questions remain to be resolved to fully benefit from AXL inhibition in the clinic.

First, AXL needs to be validated as a biomarker to stratify patients with regard to treatments, to predict responses and to prevent resistance to therapies. Given the increasing evidence of AXL implication in EMT and the deregulation of the tumor microenvironment, this RTK may be an interesting candidate for the prediction of disease progression including metastasis, relapse, and therapy response. Some studies suggest that AXL could also be an interesting marker for patients’ survival of different cancer types and a recent meta-analysis supports these observations [5,114,115]. In addition, a study using liquid biopsies for informative diagnosis in lung adenocarcinoma patients found AXL to be more highly expressed in circulating tumor cells from patients with a poor prognosis [116]. However, further large-scale analysis will be required to validate the prognostic value of AXL expression in cancer.

Secondly, since AXL inhibitors will likely enter the clinic in the near future, it will be important to predict potential secondary effects and resistance mechanisms limiting their clinical use. More fundamentally, there is a need to stratify cancers based on AXL dependency so as to select patients that could benefit the most from AXL-targeted therapies and avoid potential toxicity or resistance. Furthermore, it is key to find the right timing for the use of anti-AXL therapies in the clinic. Beyond their potential use in preventing metastases in early stages of cancer, the evidence suggesting that AXL activity can keep the disseminated tumor cells dormant could provide a way to eliminate disseminated cells by making them less resistant to other therapies. Thus, identifying the optimal timing and targets to use with anti-AXL treatment could be crucial in determining their effectiveness.

Finally, preclinical data provides ample evidence that therapeutic targeting of AXL can greatly potentiate the effectiveness of other therapies such as standard chemotherapy, radiotherapy, immunotherapy or targeted therapy. As an example, AXL inhibition could be used to prepare the field before the administration of the designed combination therapy. Accordingly, there is a great chance that the future of anti-AXL therapy in the clinic will be as a combination agent and future studies should focus on testing the most promising combinations in different contexts.

## Figures and Tables

**Figure 1 cancers-14-00466-f001:**
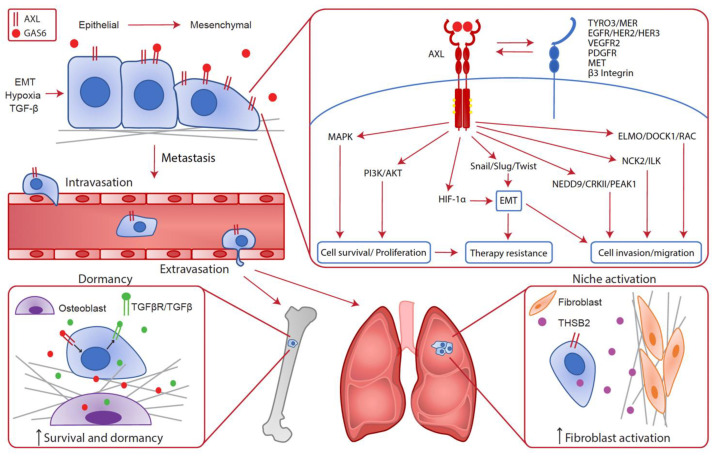
**Cancer cell intrinsic implication of AXL in the metastatic cascade.** In cancer cells, AXL expression can be enhanced by EMT, hypoxia and TGF-β leading to metastatic progression. Indeed, various cellular pathways downstream of AXL confer advantages to cancer cells such as survival, proliferation, cell invasion and migration, EMT and therapy resistance. Consequently, AXL is required throughout the metastatic cascade for local invasion, intravasation, extravasation and metastatic growth at distant sites. In the bones, AXL expression on disseminated cells increases the survival and dormancy of cancer cells by stimulating the expression of TGF-β2 and TGF-βRs. In the lungs, AXL expression in cancer cells is linked to THSB2 secretion, which activates the fibroblasts in the metastatic niche to promote metastatic growth.

**Figure 2 cancers-14-00466-f002:**
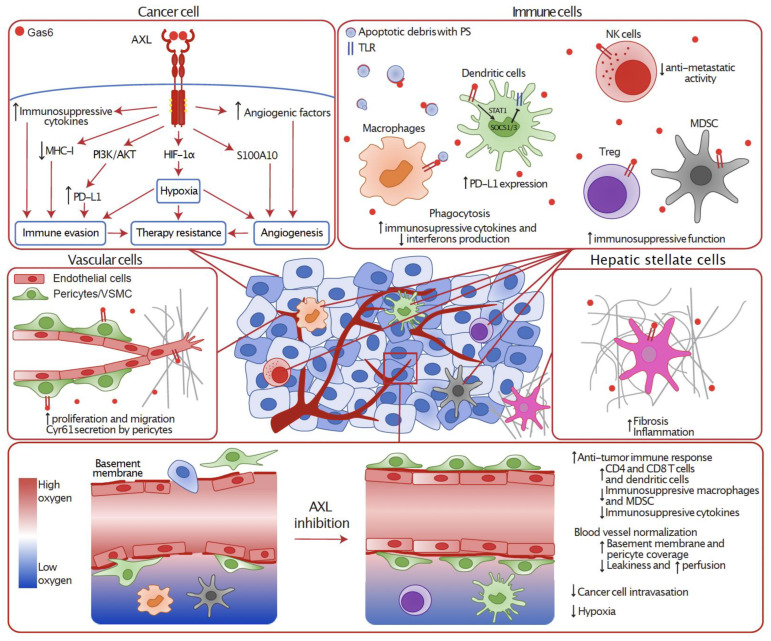
**XL implication in the tumor microenvironment in different cell types.** AXL is expressed in a variety of tumor residing cells where its activity can remodel the microenvironment. In cancer cells, AXL downstream signaling leads to hypoxia, immune evasion, angiogenesis, and therapy resistance. AXL is also expressed in different immune cells such as macrophages, dendritic cells, NK cells, regulatory T cells (Treg) and myeloid-derived suppressor cells (MDSC) where its activity leads to immunosuppression. In vascular cells, AXL promotes proliferation and migration of endothelial cells and pericytes, enhancing angiogenesis. AXL also contributes to liver fibrosis by increasing the activation of hepatic stellate cells (HSC). Thus, inhibiting AXL can remodel the tumor microenvironment by increasing the immune response and normalizing the blood vessels to reduce metastasis and improve therapy responses.

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
