# Peer review of "AXL Receptor Tyrosine Kinase as a Promising Therapeutic Target Directing Multiple Aspects of Cancer Progression and Metastasis"

_cancers, 2022, doi:10.3390/cancers14030466_

Round 1

Reviewer 1 Report

The manuscript “AXL receptor tyrosine kinase as a promising therapeutic target directing multiple aspects of cancer progression and metastasis” by Goyette and Côte, provides the discussion of AXL functions in tumor biology. Although the review is well structured and well written, the main concern is that in the present form, the review does not bring any new information to the field.

In the last decade, several review papers exploring AXL role in cancer and tumor biology as well as its implication in the development of resistance to therapy have been published Also, it is important to note that the topics presented in this manuscript are too similar to those in other AXL reviews that was previously published. In this sense, in order to be considered for publication the manuscript should be reformatted and focus in aspects that were not explored in previous AXL reviews.

Author Response

The manuscript “AXL receptor tyrosine kinase as a promising therapeutic target directing multiple aspects of cancer progression and metastasis” by Goyette and Côte, provides the discussion of AXL functions in tumor biology. Although the review is well structured and well written, the main concern is that in the present form, the review does not bring any new information to the field.

In the last decade, several review papers exploring AXL role in cancer and tumor biology as well as its implication in the development of resistance to therapy have been published. Also, it is important to note that the topics presented in this manuscript are too similar to those in other AXL reviews that was previously published. In this sense, in order to be considered for publication the manuscript should be reformatted and focus in aspects that were not explored in previous AXL reviews.

We believe our review is uniquely structured by dissecting the contributions of AXL to the metastatic cascade and the stroma. This is also how we have approach to study AXL in our recent manuscripts (Cell Reports 2018, Oncotarget 2019, Nature Communications 2020 and PNAS 2021). We do understand the point of the reviewer and after exploring again recent literature, we agree that our section on the stroma, by chance, partially overlap with a recent review on TAMs in Cancers. We have now structured our review differently to make it more unique. In particular, we separated the information based on “cancer cell intrinsic and extrinsic” cellular and molecular mechanisms. With our changes, we believe that we now make this review more unique and valuable to the broad readership of Cancers. We thank the reviewer to encourage us to do these modifications.

Reviewer 2 Report

Goyette et al provide a review of the published literature describing roles for the receptor tyrosine kinase AXL in tumor cells and the tumor microenvironment with a focus on interactions between the two compartments. The manuscript is well written and provides a useful overview of the field with a specific focus on AXL. The information is organized and clearly communicated. Relevant literature citations are included. The article is timely given the recent interest in development of TAM kinase inhibitors for treatment of cancer and their potential as immunomodulatory agents. Minor changes are suggested for clarity and completeness as indicated below.

  1. Light editing of sentence structure is needed. Eliminate the phrase “shown to be”. It doesn’t add useful information and makes statements less direct. Make sure articles (the, an, a) are used correctly.
  2. Lines 56-58: Should soften the language around heterodimerization of AXL with other family members. The study cited shows a functional interaction. Other published studies showing co-immunoprecipitation should be cited. These studies all suggest heterodimerization. More extensive experiments would be needed to demonstrate heterodimerization.
  3. Lines 60-62: The statement that there’s a negative correlation of GAS6 expression with survival in breast cancer is confusing. A negative correlation suggests an increase in one is associated with a decrease in the other. Here, increased GAS6 correlates with increased survival, which is a direct correlation. State this more explicitly.
  4. In Figure 2, some panels show the effects of AXL activation (macrophages and dendritic cells, cancer cells, vascular cells, Treg and MDSC), but the NK cell panel shows the effects of AXL inhibition (decreased metastasis). NK panel should be changed to be consistent with the other panels.
  5. Lines 197-199: States “Many studies have shown a better drug sensitivity when combining AXL inhibition with chemotherapeutic compounds such as Docetaxel, Cisplatin, Pemetrexel, Vincristine, Paclitaxel, Adryamicin or Gemcitabine [65-68]”. Add carboplatin to the list and cite https://pubmed.ncbi.nlm.nih.gov/22890323/.
  6. Section 3.4: Expand this section to indicate roles for AXL in resistance to a wide variety of targeted agents (osimertinib, venetoclax, ATR inhibitor, FGFR inhibitor, IGF1R inhibitor, MERTK inhibition, FLT3 inhibitor).
  7. Lines 249-251: States “Several studies now point out that the TAMs, and particularly AXL, contribute to an immunosuppressive tumor microenvironment and can modulate the ability of immune cells to eliminate cancer cells”. This sentence gives the impression that AXL plays a more prominent role than the other TAM kinases and should be rephrased as “Several studies now point out that AXL contributes to an immunosuppressive tumor microenvironment and can modulate the ability of immune cells to eliminate cancer cells”.
  8. Lines 254-255: States “…AXL activation can dampen the immune response against tumor cells and reprogramme the infiltrated immune cells to promote their proliferation and invasion”. Does “their proliferation and invasion” refer to tumor cells or immune cells? Needs clarification.
  9. Lines 261-263: States that AXL regulates PD-L1 expression. Add the following citation: https://www.ncbi.nlm.nih.gov/pmc/articles/PMC8363069/pdf/nihms847610.pdf.
  10. In section, 4.1, reorganize subsections to talk about the related subjects of angiogenesis and hypoxia, then followed by immune microenvironment.
  11. Section 4.3 should mention the therapeutic implications of direct tumor cell targeting. For instance, add a sentence to state that inhibition of AXL in tumor cells is expected to enhance sensitivity and/or prevent resistance to cytotoxic chemotherapies and targeted agents, such as EGFR tyrosine kinase inhibitors, providing rationale for application of AXL-targeted agents in combination with other therapies.

Author Response

Goyette et al provide a review of the published literature describing roles for the receptor tyrosine kinase AXL in tumor cells and the tumor microenvironment with a focus on interactions between the two compartments. The manuscript is well written and provides a useful overview of the field with a specific focus on AXL. The information is organized and clearly communicated. Relevant literature citations are included. The article is timely given the recent interest in development of TAM kinase inhibitors for treatment of cancer and their potential as immunomodulatory agents. Minor changes are suggested for clarity and completeness as indicated below.

We would like to thank reviewer #2 for his interest in our manuscript.

  1. Light editing of sentence structure is needed. Eliminate the phrase “shown to be”. It doesn’t add useful information and makes statements less direct. Make sure articles (the, an, a) are used correctly.

We requested the help of a professional editor to assist us in improving the English language in our manuscript.

  1. Lines 56-58: Should soften the language around heterodimerization of AXL with other family members. The study cited shows a functional interaction. Other published studies showing co-immunoprecipitation should be cited. These studies all suggest heterodimerization. More extensive experiments would be needed to demonstrate heterodimerization.

This is an excellent comment. Accordingly, we soften the information dealing with heterodimerization of RTKs and included additional references on this topic.

  1. Lines 60-62: The statement that there’s a negative correlation of GAS6 expression with survival in breast cancer is confusing. A negative correlation suggests an increase in one is associated with a decrease in the other. Here, increased GAS6 correlates with increased survival, which is a direct correlation. State this more explicitly.

We revised this section based on this comment and we believe the message to be clearer.

  1. In Figure 2, some panels show the effects of AXL activation (macrophages and dendritic cells, cancer cells, vascular cells, Treg and MDSC), but the NK cell panel shows the effects of AXL inhibition (decreased metastasis). NK panel should be changed to be consistent with the other panels.

As requested, Figure 2 was modified to be more consistent across the figure panels.

Lines 197-199: States “Many studies have shown a better drug sensitivity when combining AXL inhibition with chemotherapeutic compounds such as Docetaxel, Cisplatin, Pemetrexel, Vincristine, Paclitaxel, Adryamicin or Gemcitabine [65-68]”. Add carboplatin to the list and cite https://pubmed.ncbi.nlm.nih.gov/22890323/.

Thank you, this reference to carboplatin is now included in that section.

  1. Section 3.4: Expand this section to indicate roles for AXL in resistance to a wide variety of targeted agents (osimertinib, venetoclax, ATR inhibitor, FGFR inhibitor, IGF1R inhibitor, MERTK inhibition, FLT3 inhibitor).

For this section, we decided to concentrate on the resistance caused by EMT induction, since it was the focus of the section. Several reviews already tackled this subject in dept, so we decided not to elaborate to avoid redundancy, but we added more references to acknowledge these studies in section 2.1. We hope the reviewer will be satisfied by our approach.

  1. Lines 249-251: States “Several studies now point out that the TAMs, and particularly AXL, contribute to an immunosuppressive tumor microenvironment and can modulate the ability of immune cells to eliminate cancer cells”. This sentence gives the impression that AXL plays a more prominent role than the other TAM kinases and should be rephrased as “Several studies now point out that AXL contributes to an immunosuppressive tumor microenvironment and can modulate the ability of immune cells to eliminate cancer cells”.

We edited the sentence according to this excellent comment.

  1. Lines 254-255: States “…AXL activation can dampen the immune response against tumor cells and reprogramme the infiltrated immune cells to promote their proliferation and invasion”. Does “their proliferation and invasion” refer to tumor cells or immune cells? Needs clarification.

The text was restructured and this information was written in a different way, so this important comment is not relevant anymore.  

  1. Lines 261-263: States that AXL regulates PD-L1 expression. Add the following citation: https://www.ncbi.nlm.nih.gov/pmc/articles/PMC8363069/pdf/nihms847610.pdf.

Thank you, this reference is now included in this section.

  1. In section, 4.1, reorganize subsections to talk about the related subjects of angiogenesis and hypoxia, then followed by immune microenvironment.

We decided to keep the same order in the text since hypoxia is linked to angiogenesis and immune response, so it was making more sense to put hypoxia after the others.

  1. Section 4.3 should mention the therapeutic implications of direct tumor cell targeting. For instance, add a sentence to state that inhibition of AXL in tumor cells is expected to enhance sensitivity and/or prevent resistance to cytotoxic chemotherapies and targeted agents, such as EGFR tyrosine kinase inhibitors, providing rationale for application of AXL-targeted agents in combination with other therapies.

The addition to section 3.4 now covers this excellent point.

Reviewer 3 Report

The authors present a succinct review on the rationale behind using AXL receptor tyrosine kinase as a target in cancer. The manuscript focuses on the biological aspects of AXL function and its role in cancer, rather than reviewing the more clinical aspects, as these have been the subject of several recent reviews. However, there are many recent reviews addressing this topic or focusing on specific cancer types. In PubMed, the search for reviews of "AXL" AND "cancer" gave 169 results, 18 with these words in the title, six of them since 2019. Leaving aside the subject of novelty, the review presented by Goyette and Cote is well written and provides an extensive description of the literature relevant to the topic. My only suggestion would be to include also the link between AXL and tissue fibrosis. This aspect has been extensively reported in the literature and it is of paramount importance in the case of hepatic cancer, linking NAFL/NASH with Hepatocellular carcinoma. As the review deals with biological processes where AXL is involved and impacting cancer development, I suggest including a section summarizing the AXL-fibrosis connection.

Both figures are nice captions describing the effects of AXL. However, the legend should be more explanatory, including a description of each section of the figure.

Although the text is well written and clear, I have some specific suggestions.

Line 95:  "...the implications of AXL in a variety of cancer related biological processes that are tumor cells intrinsic and extrinsic." The sentence is not clear to me. Maybe the authors could expand on what do they consider intrinsic and extrinsic in this context.

Line 101: "...give the cancer cell..." (or "cancer cells")

Line 136: "TNBC cells" Could the authors provide the definition of the acronym? Alternatively, they could include it in "In triple-negative breast cancer (TNBC)..." in line 87.

Line 158: "...in epithelial cell lines"

Line 164: "confirm"

Line 227: "require the acquisition"

Line 240: "in vitro and in vivo"

Line 303: "primarily"

Author Response

The authors present a succinct review on the rationale behind using AXL receptor tyrosine kinase as a target in cancer. The manuscript focuses on the biological aspects of AXL function and its role in cancer, rather than reviewing the more clinical aspects, as these have been the subject of several recent reviews. However, there are many recent reviews addressing this topic or focusing on specific cancer types. In PubMed, the search for reviews of "AXL" AND "cancer" gave 169 results, 18 with these words in the title, six of them since 2019. Leaving aside the subject of novelty, the review presented by Goyette and Cote is well written and provides an extensive description of the literature relevant to the topic. My only suggestion would be to include also the link between AXL and tissue fibrosis. This aspect has been extensively reported in the literature and it is of paramount importance in the case of hepatic cancer, linking NAFL/NASH with Hepatocellular carcinoma. As the review deals with biological processes where AXL is involved and impacting cancer development, I suggest including a section summarizing the AXL-fibrosis connection.

Thank you for the suggestion, a section on fibrosis was added to the text and Figure 2.

Both figures are nice captions describing the effects of AXL. However, the legend should be more explanatory, including a description of each section of the figure.

The legends were modified to include more details.

Although the text is well written and clear, I have some specific suggestions.

Line 95:  "...the implications of AXL in a variety of cancer related biological processes that are tumor cells intrinsic and extrinsic." The sentence is not clear to me. Maybe the authors could expand on what do they consider intrinsic and extrinsic in this context.

Line 101: "...give the cancer cell..." (or "cancer cells")

The editing was done, thank you. We also secure the help of a professional writer to globally improve the clarity of our manuscript.

Line 136: "TNBC cells" Could the authors provide the definition of the acronym? Alternatively, they could include it in "In triple-negative breast cancer (TNBC)..." in line 87.

We forgot to include the definition, thank you for pointing it out.

Line 158: "...in epithelial cell lines"

Line 164: "confirm"

Line 227: "require the acquisition"

Line 240: "in vitro and in vivo"

Line 303: "primarily"

All these typos were corrected.

Reviewer 4 Report

In the present review, Goyette and Coté discuss the role of AXL in cancer cells and its possible use as a therapeutic target.
The work is well-conceived, MoA of AXL is described as well as its role in metastasis and resistance to therapy. This leads the readers to clearly understand the advantages of targeting this protein. Moreover, the authors discuss the role of AXL in tumor microenvironment and anti-tumor immunity which are interesting hot topics. 
This review is clear and interesting and has only minor issues regarding language and syntax.

Author Response

In the present review, Goyette and Coté discuss the role of AXL in cancer cells and its possible use as a therapeutic target.
The work is well-conceived, MoA of AXL is described as well as its role in metastasis and resistance to therapy. This leads the readers to clearly understand the advantages of targeting this protein. Moreover, the authors discuss the role of AXL in tumor microenvironment and anti-tumor immunity which are interesting hot topics. 
This review is clear and interesting and has only minor issues regarding language and syntax.

Thank you for the positive comments, we edited the text with the help of a professional writer to improve the use of the English language and the clarity of our messages.

Round 2

Reviewer 1 Report

The modifications performed highly improved the manuscript quality. Although, as said in the previous comments, several reviews in Axl role in cancer progression, metastasis and as therapeutic target are available, however the present review is well written and structured and will be of use for those in the field.